# Endoscopic Enucleation of Prostate Could Increase Testosterone Levels in Hypotestosteronemic Patients with Bladder Outlet Obstruction

**DOI:** 10.3390/jcm11226808

**Published:** 2022-11-17

**Authors:** Yun-Ren Li, Shu-Han Tsao, Chien-Lun Chen, Chen-Pang Hou, Ke-Hung Tsui, Horng-Heng Juang, Yu-Hsiang Lin

**Affiliations:** 1Department of Urology, Chang Gung Memorial Hospital-Linkou, Kwei-Shan, Taoyuan 33302, Taiwan; 2School of Medicine, Chang Gung University, Kwei-Shan, Taoyuan 33302, Taiwan; 3Department of Urology, Shuang Ho Hospital, Taipei Medical University, Taipei 23541, Taiwan; 4TMU Research Center of Urology and Kidney (TMU-RCUK), Taipei Medical University, Taipei 11031, Taiwan; 5Department of Anatomy, College of Medicine, Chang Gung University, Kwei-Shan, Taoyuan 33302, Taiwan

**Keywords:** benign prostatic hyperplasia, secondary hypogonadism, testosterone, circadian rhythm, BipolEP, ThuLEP

## Abstract

Background: We evaluated the impact of endoscopic enucleation of the prostate on testosterone levels in hypotestosteronemic patients with bladder outlet obstruction. Methods: We enrolled 294 men with lower urinary tract symptoms (LUTS) who received surgery between January 2019 and December 2020 in simple tertiary centre. The inclusion criteria were as follows: being a male patient aged 45–95 years and having recurrent urinary tract infection, having previously failed medical treatment for LUTS or urine retention, and undergoing bipolar or thulium laser enucleation of the prostate. The preoperative and postoperative data were retrospectively reviewed. Results: This study included 112 men with a mean age of 69.4 years. The mean preoperative and postoperative testosterone levels were 4.8 and 4.98, respectively. Of the patients, 88 (78.6%) received ThuLEP and 24 received BipolEP. We divided the patients into two groups according to preoperative serum testosterone levels: normal-testosterone (≥3 ng/mL) and low-testosterone (<3 ng/mL) groups. A significant change in testosterone levels (*p* = 0.025) was observed in the low-testosterone group. In contrast, no significant difference in testosterone levels was noted in the normal-testosterone group (*p* = 0.698). Conclusions: Endoscopic enucleation surgery of the prostate could improve postoperative testosterone levels in hypotestosteronemic patients with bladder outlet obstruction.

## 1. Introduction

Benign prostatic obstruction (BPO) or benign prostatic enlargement (BPE) is characterized by an increased number of epithelial and stromal cells in the periurethral area of the prostate, causing a spectrum of lower urinary tract symptoms (LUTS), namely, frequency, urgency, nocturia, weak stream, intermittency, straining to void, and incomplete bladder emptying. LUTS associated with BPO or BPE can exert a considerable negative effect on patients’ quality of life (QoL), prompting them to seek treatment [1]. However, BPE is considered a natural consequence of aging. Approximately 70% of men aged 60–69 years in the United States have a certain degree of BPO [2]. Similar to menopause in women, late-onset hypogonadism (LOH) in men is a clinical condition associated with low serum sex hormone levels; its clinical symptoms are decreased libido, bone mineral density, and skeletal muscle mass and increased visceral fat, anemia, amnesia, and even insulin resistance [3,4]. A review study indicated that low testosterone levels are associated with increased risks of diabetes mellitus and cardiovascular disease [4]. However, testosterone supplements are not widely applied in patients with hypotestosteronemia because of the associated risks—the most notable of which is the possible stimulation of prostate cancer by testosterone—although strong evidence supporting such risks is not available [5]. However, other studies have demonstrated improvement in serum testosterone levels in patients with BPO or BPE after silodosin [6] and vasopressin [7] administration to relieve LUTS, especially nocturia. This implies that serum testosterone could be improved with a better sleep and circadian rhythm [8]. Nevertheless, this topic warrants further research. Accordingly, we conducted this retrospective study to investigate the changes in testosterone levels after surgical intervention in patients with BPO or BPE which may rescue patients’ sleep and circadian rhythm by decreasing nocturia.

## 2. Materials and Methods

We enrolled 294 men with LUTS who received prostate enucleation between January 2019 and December 2020 at Linkou Chang Gung Memorial Hospital. The study protocol was approved by the Institutional Review Board (IRB) of Chang Gung Memorial Hospital, Taoyuan, Taiwan (IRB: 202001392B0). Due to the retrospective nature of this study, the IRB waived the requirement for obtaining patients’ consent to review their medical records. Patient data confidentiality was protected in accordance with the Declaration of Helsinki. The study inclusion criteria were as follows: being a male patient aged 45–95 years and having recurrent urinary tract infection, having previously failed medical treatment for LUTS or urine retention, and undergoing bipolar enucleation of the prostate (BipolEP) or thulium laser enucleation of the prostate (ThuLEP) for the surgical relief of prostatic obstruction by a single experienced surgeon. Patients were excluded if the histopathology results confirmed the diagnosis of prostate cancer.

For all patients, a comprehensive history-taking, physical examination, transrectal sonography, uroflowmetry, complete blood count, serum electrolyte, creatinine, prostate-specific antigen (PSA), serum testosterone, and urine analysis were performed preoperatively. The serum testosterone level was obtained in the morning because of the circadian rhythm. At 1.5 months after surgery, the hormonal assessment was performed again. The resection ratio was defined as the BipolEP or ThuLEP specimen weight divided by the total prostatic volume (measured through transrectal echography). The adverse events with ER visit including urinary tract infection, bleeding with surgical intervention, urethral stricture and recatheterization were recorded during the post-operative 3 months follow-up.

For statistical analysis, categorical and continuous variables were analysed using chi-squared test and independent *t* test. Moreover, variables for which the *p* value was <0.05 in the univariate analysis were included in a multivariate linear regression model for further analysis. The paired *t* test was used to assess differences in testosterone levels before and after surgery. All reported *p* values were single-sided with a statistical significance of <0.05. In addition, all statistical analyses were performed using IBM SPSS Statistics 22 software.

## 3. Results

We initially enrolled 294 patients in this study. Of these patients, 88 did not have preoperative testosterone data, 66 did not obtained a morning blood sample to evaluate testosterone level, and 28 had histopathology results confirming the diagnosis of prostate cancer and were thus excluded. Thus, the clinical data of the remaining 112 patients with a mean age of 69.4 years were analysed (Table 1 and Table 2). The average body height and mean body weight of the patients were 166.3 cm and 69.8 kg, respectively, and the derived average body mass index (BMI) was 25.2 kg/m^2^. Personal and medical history records revealed that of the patients, 21.4% had diabetes mellitus, 54.5% had hypertension, 10.7% had a stroke history, 1.8% had Parkinson’s disease, and 15.2% underwent anticoagulant therapy. The average PSA level was 4.34, and the mean preoperative and postoperative testosterone levels were 4.80 and 4.98, respectively. The average preoperative and postoperative nocturia episodes was 3.3 and 1.6, respectively. Moreover, of the patients, 88 (78.6%) received ThuLEP and 24 underwent BipolEP, with the average resection ratio being 51.5%. The adverse events with ER were as follows: bleeding with surgical intervention in 2 patients, urinary tract infection with oral antibiotics treatment in 2 patients, urinary retention with recatheterization in 1 patients, and urethral stricture which received further surgery in 3 patients.

The paired *t* test results regarding the preoperative and postoperative testosterone levels are presented in Table 3. We noted that the testosterone level was not significantly elevated in response to treatment in the entire cohort. We further divided the patients into two groups according to preoperative serum testosterone levels: normal-testosterone (≥3 ng/mL) and low-testosterone (<3 ng/mL) groups. The changes in preoperative postoperative testosterone levels are presented in Figure 1. We observed a significant change in testosterone levels (from 2.26 to 2.99; *p* = 0.025) in the low-testosterone group. In contrast, we observed no significant difference in testosterone levels (5. 34 to 5.41; *p* = 0.698) in the normal-testosterone group.

The paired *t* test results regarding the changes in nocturia episodes during preoperative and postoperative periods are presented in Table 4. We noted that the nocturia episodes were significantly improved in response to treatment in the entire cohort (from 3.3 to 1.6; *p* < 0.01), the low-testosterone group (from 3.2 to 1.9; *p* < 0.01) and the normal-testosterone group (from 3.3 to 1.6; *p* < 0.01).

Furthermore, we analysed factors that may contribute to low preoperative testosterone levels in patients who received surgery for BPE. Multivariable logistic regression revealed that diabetes mellitus (odds ratio [OR] = 4.718; *p* = 0.02) and a high BMI (OR = 1.226; *p* = 0.03) tended to contribute to relatively low testosterone levels (Table 5).

## 4. Discussion

The relationship between testosterone and BPO or BPE remains unclear. Rohrmann et al., noted that BPO or BPE is significantly associated with metabolic syndrome and shares the same abnormalities with metabolic syndrome [9]. McVary et al., revealed similarities between erectile dysfunction and LUTS: metabolic syndrome, autonomic nervous activity, nitric oxide activity, arteriosclerosis, pelvic ischemia, and Rho-kinase activity [10]. Moreover, patients with diabetes mellitus were reported to have poorer transurethral prostatic resection (TURP) outcomes than those patients without diabetes mellitus [11]. Tan et al., also reported that a decrease in serum free testosterone concentrations in advanced age may be a major factor in the development of BPH in men aged 60 to 69 years [12]. These findings imply the relationship between decreased testosterone levels and BPE or BPO.

Studies have extensively investigated the association between obesity and type-2-diabetes-mellitus-related secondary hypogonadism, also known as male obesity-associated secondary hypogonadism (MOSH). The mechanisms implicated in the association of male secondary hypogonadism with obesity, type 2 diabetes mellitus, and metabolic features are complex [13]. Male obesity and type 2 diabetes mellitus are associated with increased aromatase activity in adipocytes, resulting in increased serum oestradiol levels [14]. Therefore, oestradiol, suppresses the hypothalamic–pituitary–gonadal axis, leading to a reduction in plasma testosterone levels and MOSH [15]. Furthermore, an obesity-induced increase in leptin, insulin, and proinflammatory cytokine levels can cause hypogonadism [16]. Although the pathogenesis of MOSH remains unclear, studies have demonstrated that male obesity is essentially associated with lower plasma testosterone levels [17], and men with obesity and type 2 diabetes mellitus appear to be at a particularly high risk of MOSH [18,19]. These findings are consistent with our results, which reveal that high BMI and type 2 diabetes mellitus were independent predictive factors for low preoperative testosterone levels.

This study is the first to evaluate changes in total serum testosterone levels in patients undergoing BipolEP and ThuLEP for the surgical relief of prostatic obstruction. Türkölmez et al., demonstrated preoperative and postoperative differences in hormone levels in 57 patients with BPH who received TURP or transurethral laser prostatectomy (TULP). In the patients who received TURP, the LH and prolactin levels were significantly higher at the 3-week postoperative time point than those at the preoperative time point, but the levels observed at the 3-month postoperative time point did not differ from those observed preoperatively. In the patients who received TULP, only the plasma LH levels were significantly increased at the 3-month time point relative to the preoperative time point. Therefore, Kadir Türkölmez et al., considered that some factors released from the resected prostate gland may have affected the evaluation results and that the differences may be related to a higher amount of residual prostate tissue after TULP [20]. Furthermore, previous observational studies have reported no changes in testosterone, LH, FSH, adrenocorticotropic hormone, or dehydroepiandrosterone sulfate levels at short-term (3 weeks) [21] and 3-month follow-up time points after surgery [20]. In the present study, we noted a significant difference between preoperative and postoperative testosterone levels in the low-testosterone group (*p* < 0.05). This difference may be due to the high resection ratio during enucleation surgery in our study (the average resection ratio was 51.5% of the total prostatic volume). In the studies by Türkölmez et al. [20] and Sasagawa et al. [21], the cohorts were not divided into normal and hypotestosteronemic subgroups to evaluate the effect of surgery on hormone levels; this may echo our conclusion androgen level elevation may exist especially in hypotestosteronemic patients whose sleep was interrupted by nocturia.

Serum testosterone exhibits a circadian rhythm in adult men. Moreover, the mean testosterone levels are lower in healthy older men than in younger men. The circulatory testosterone concentrations peak in the morning and decrease during the day. Therefore, any disturbance in sleep may reduce the nocturnal increase in testosterone levels [22,23], such as BPE- or BPO-related nocturia. Kaufman et al., revealed that the circadian rhythm of serum testosterone in normal young men was markedly attenuated or absent in healthy older men [24,25]. Therefore, the characteristic early morning rise in testosterone levels in young men would not be present in old age. Matsukawa et al., conducted a prospective study including patients with BPO; they revealed that after 1 year of silodosin administration, the mean serum testosterone levels increased significantly (5.09 to 5.52 ng/mL), and they observed a significant positive correlation between the change in testosterone levels and improvement in the bladder outlet obstruction index [6]. Additionally, Kim et al., reported that using desmopressin as a treatment for nocturnal polyuria in men with LOH syndrome significantly increased testosterone levels [7]. Once a patient receives medication or surgical intervention to relieve the nocturia problem, the circadian rhythm might be restored, causing the elevation of postoperative testosterone levels. Our previous study revealed that, in addition to testosterone restoration, TURP provides favourable clinical outcomes such as reduced UTI and urine retention incidence and reduced life-long bone fracture in patients with BPO-induced urine retention [26]. In the present study, decreased nocturia episodes were noted in the whole cohort, but only the low-testosterone group showed the elevation of the serum testosterone levels especially in the low-testosterone group (2.23 to 2.91 ng/mL) at 1.5 months after surgery. This may imply the restoration of circadian rhythm of testosterone was related to less sleep interruptions by decreasing nocturia episodes. Although the testosterone elevation only showed up in the low-testosterone group at 1.5 months, longer follow-up periods may be needed to confirm the trend in serum testosterone levels in these patients. Furthermore, future prospective studies must be performed to confirm the hypothesis that circadian rhythm restoration can improve hypotestosteronemia in patients with BPO or BPE.

This was a retrospective cohort study with a medium sample size; therefore, it has several limitations. First, surgery was performed by a single surgeon. Second, we did not collect complete sex hormone profiles of the patients. Finally, the lack of information on IPSS (International Prostatism Symptom Score), hypogonadism symptoms and QoL may reduce the strength of our findings regarding postoperative outcomes. Therefore, further research is necessary to overcome these limitations.

## 5. Conclusions

Endoscopic enucleation surgery of the prostate could improve postoperative testosterone levels in hypotestosteronemic patients with bladder outlet obstruction.

## Figures and Tables

**Figure 1 jcm-11-06808-f001:**
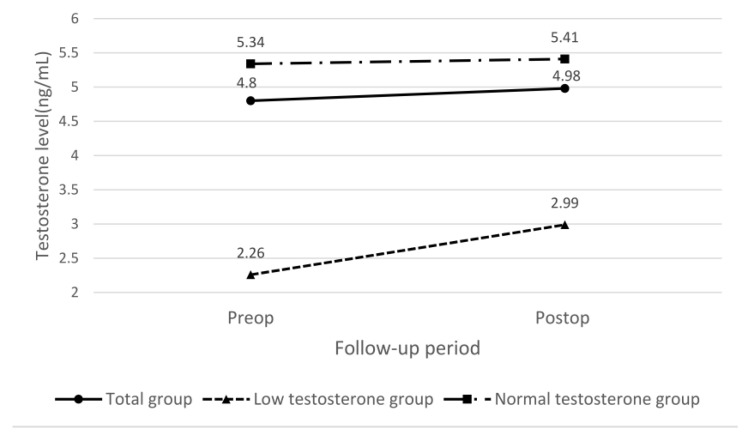
Preoperative and postoperative testosterone levels in the low- and normal-testosterone groups.

**Table 1 jcm-11-06808-t001:** Demographic data of the low- and normal-testosterone groups.

	Total	Low-Testosterone Group (20)	Normal-Testosterone Group (92)	*p* Value
Variable	Mean ± SD or *n* (%)			
Age(year)	69.4 ± 8.9	69.6 ± 10.8	69.4 ± 8.5	0.950
Height(cm)	166.3 ± 5.5	166.8 ± 5.9	166.2 ± 5.4	0.662
Weight(kg)	69.8 ± 10.5	75.7 ± 13.4	68.5 ± 9.3	<0.01
BMI (kg/m^2^)	25.2 ± 3.4	27.3 ± 4.7	24.8 ± 2.9	<0.01
Diabetes mellitus	24 (21.4)	9 (45)	15 (16.3)	<0.01
Hypertension	61 (54.5)	14 (70)	47 (51.1)	0.097
Stroke history	12 (10.7)	2 (10)	10 (10.9)	0.889
Parkinson disease	2 (1.8)	1 (5)	1 (1.1)	0.441
Under anticoagulant therapy	17 (15.2)	5 (25)	12 (13)	0.156
Preop PSA (ng/mL)	4.34 ± 4.53	3.68 ± 2.44	4.48 ± 4.86	0.477
Preop testosterone (ng/mL)	4.80 ± 1.81	2.26 ± 0.61	5.35 ± 1.48	<0.01
Preop nocturia	3.3 ± 1.7	3.2 ± 1.7	3.3 ± 1.7	0.712

**Table 2 jcm-11-06808-t002:** Postoperative differences between the low- and normal-testosterone groups.

	Total	Low-Testosterone Group	Normal-Testosterone Group	*p* Value
Variable	Mean ± SD or *n* (%)			
Postop PSA (ng/mL)	1.52 ± 1.47	1.34 ± 1.16	1.56 ± 1.53	0.554
Postop Testosterone (ng/mL)	4.98 ± 1.86	2.99 ± 1.46	5.41 ± 1.65	<0.01
Postop nocturia	1.6 ± 1.0	1.9 ± 1.0	1.6 ± 1.0	0.914
Prostate volume (mL)	51.9 ± 33.6	47.1 ± 18.2	52.8 ± 36.1	0.491
Specimen weight (gm)	30.7 ± 37.7	18.6 ± 17.5	33.3 ± 40.3	0.114
Resection ratio (%)	51.5 ± 35.3	34.7 ± 27.4	55.1 ± 35.9	0.019
Adverse events with ER visit A-blocker discontinuation Ratio (%)	10 (8.9) 106 (94.6)	2 (10) 20 (100)	8 (8.7) 86 (93.5)	0.566 0.298

**Table 3 jcm-11-06808-t003:** Comparison between preoperative and postoperative testosterone levels in the low- and normal-testosterone groups.

	Preop Testosterone	Postop Testosterone		
Group	Mean ± SD	Mean ± SD	t	*p* Value
Total (112)	4.80 ± 1.81	4.98 ± 1.86	−1.21	0.229
Low-testosterone (20)	2.26 ± 0.61	2.99 ± 1.46	−2.442	0.025
Normal-testosterone (92)	5.35 ± 1.48	5.41 ± 1.65	−0.39	0.698

Low-testosterone group: preoperative testosterone < 3 ng/mL; normal-testosterone group: preoperative testosterone ≥ 3 ng/mL.

**Table 4 jcm-11-06808-t004:** Comparison of the frequency of preoperative and postoperative nocturia episodes in the low- and normal-testosterone groups.

	Preop Nocturia	Postop Nocturia		
Group	Mean ± SD	Mean ± SD	t	*p* Value
Total (112)	3.3 ± 1.7	1.6 ± 1.0	8.623	<0.01
Low-testosterone (20)	3.2 ± 1.7	1.9 ± 1.0	3.053	<0.01
Normal-testosterone (92)	3.3 ± 1.7	1.6 ± 1.0	8.073	<0.01

**Table 5 jcm-11-06808-t005:** Multivariable logistic regression analysis of factors predicting low preoperative testosterone levels.

Variable	Odds Ratio	95% CI	*p*
Age	0.97	0.900–1.044	0.412
Diabetes mellitus	4.718	1.254–17.757	0.02
Hypertension	1.359	0.409–4.508	0.617
Anticoagulant therapy	1.38	0.304–6.273	0.677
Stroke history	0.744	0.101–5.466	0.772
Parkinson disease	2.591	0.073–91.57	0.601
BMI	1.226	1.020–1.475	0.030

BMI = body mass index.

## Data Availability

Not applicable.

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
