# Peer review of "Endoscopic Enucleation of Prostate Could Increase Testosterone Levels in Hypotestosteronemic Patients with Bladder Outlet Obstruction"

_jcm, 2022, doi:10.3390/jcm11226808_

Round 1
Reviewer 1 Report
The aim of the study was to evaluate the impact the impact of endoscopic enucleation of the prostate on testosterone levels in hypotestosteronemic patients with bladder outlet obstruction. Topic could be interesting even if there are several critical aspects which undermine the overall quality of manuscript.
- Preoperative drugs should be described in order to evaluate the impact of dutasteride and finasteride on testosterone levels.
- Authors reported in low testosterone group a significant increase in postoperative testosterone levels, reporting a mean postoperative tesosterone of 2.99. However, this increase seems to be clinically insignificant bening <3, which is the cut off using to divide patients in low and normal testosterone. Therefore, the significant increase postoperatively could not be clinically significant.
Author Response
Reviewer 1
The aim of the study was to evaluate the impact the impact of endoscopic enucleation of the prostate on testosterone levels in hypotestosteronemic patients with bladder outlet obstruction. Topic could be interesting even if there are several critical aspects which undermine the overall quality of manuscript.
Q1.- Preoperative drugs should be described in order to evaluate the impact of dutasteride and finasteride on testosterone levels.
Thanks for your comment. In designing this retrospective study, we do not notice this issue. After reviewing the literature, we noted that the association between dutasteride/finasteride prescription and serum testosterone varies. Most series shows elevated serum testosterone level[1-3] but there would be some shows decreased[4] or relative decreased testosterone level to placebo arm[5]. It seems no definite conclusion about the effect of dutasteride/finasteride on serum testosterone. And we prescribed 5-alfa-reductase inhibitors (5ARI) passively for the fear of partial androgen deprivation effect and related metabolic syndrome[6]. As a result, there are supposed to have rare patients receiving 5ARI in our cohort. And after reviewing our cohort (112 pts ), there is no patient receiving 5ARI.
This manuscript focuses on the effect of testosterone after prostate enucleation. In the current literature, 5ARI has no exact effect on serum testosterone and we have no patient recieving 5ARI in this analyzed cohort. As a result, we do not change our manuscript on this issue.
Q2. Authors reported in low testosterone group a significant increase in postoperative testosterone levels, reporting a mean postoperative tesosterone of 2.99. However, this increase seems to be clinically insignificant bening <3, which is the cut off using to divide patients in low and normal testosterone. Therefore, the significant increase postoperatively could not be clinically significant.
-àThanks for your comment. Certainly, testosterone level does not reach 3.0 in this cohort. But it indeed increases in the pre-op low testosterone group after prostate enucleation. And we raise this study owing to that we notice our enucleation patients are spiritual clinically during postop follow up. It is a pity that due to the retrospective nature of this study, we don’t have “Androgen Deficiency in Aging Males (ADAM) questionnaire” before.
Thanks for your precious suggestion, we have changed our title from “Restore Testosterone Levels” to “Increase Testosterone Levels” for not reaching 3.
[1] Clark RV, Hermann DJ, Cunningham GR, Wilson TH, Morrill BB, Hobbs S. Marked suppression of dihydrotestosterone in men with benign prostatic hyperplasia by dutasteride, a dual 5α-reductase inhibitor. The journal of clinical endocrinology & metabolism. 2004;89:2179-84.
[2] Hong SK, Min GE, Ha SB, Doo SH, Kang MY, Park HJ, et al. Effect of the dual 5α‐reductase inhibitor, dutasteride, on serum testosterone and body mass index in men with benign prostatic hyperplasia. BJU international. 2010;105:970-4.
[3] Olsen EA, Hordinsky M, Whiting D, Stough D, Hobbs S, Ellis ML, et al. The importance of dual 5α-reductase inhibition in the treatment of male pattern hair loss: results of a randomized placebo-controlled study of dutasteride versus finasteride. Journal of the American Academy of Dermatology. 2006;55:1014-23.
[4] Traish AM, Haider KS, Doros G, Haider A. Finasteride, not tamsulosin, increases severity of erectile dysfunction and decreases testosterone levels in men with benign prostatic hyperplasia. Hormone molecular biology and clinical investigation. 2015;23:85-96.
[5] Wurzel R, Ray P, Major-Walker K, Shannon J, Rittmaster R. The effect of dutasteride on intraprostatic dihydrotestosterone concentrations in men with benign prostatic hyperplasia. Prostate Cancer and Prostatic Diseases. 2007;10:149-54.
[6] Bianchi V, Locatelli V. Testosterone a key factor in gender related metabolic syndrome. Obesity Reviews. 2018;19:557-75.

Reviewer 2 Report
1) General comments
The authors aim to evaluate the impact of endoscopic enucleation of the prostate on testosterone levels in hypotestosteronemic patients with bladder outlet obstruction. They showed a significant change in testosterone levels (p=0.025) was observed in the low testosterone group. By contrast, no significant difference in testosterone levels
was noted in the normal-testosterone group (p = 0.698).
They concluded that endoscopic enucleateion surgery of the prostate could improve postoperative testosterone levels in hypotestosteronemic patients with bladder outlet obstruction.
The reviewer generally agrees with the conclusion.
However, there are several issues need to improve. The reviewer would like suggests several issues as follows;
2) Specific comments for revision
a) Major
#1 Are there any reports of testosterone changes in other enucleation procedures such as HoLEP?
#2 How much did the frequency of nocturnal urination change before and after surgery in the patient group in this study? Is it possible that the decrease in nocturnal urination and the improved sleep environment may have increased postoperative testosterone?
b) Minor
#1 Low testosterone groups tended to have less prostatic nucleation weight. Is there any report that the larger the prostate volume, the lower the testosterone level?
Author Response
Reviewer 2
1) General comments
The authors aim to evaluate the impact of endoscopic enucleation of the prostate on testosterone levels in hypotestosteronemic patients with bladder outlet obstruction. They showed a significant change in testosterone levels (p=0.025) was observed in the low testosterone group. By contrast, no significant difference in testosterone levels was noted in the normal-testosterone group (p = 0.698).
They concluded that endoscopic enucleateion surgery of the prostate could improve postoperative testosterone levels in hypotestosteronemic patients with bladder outlet obstruction.
The reviewer generally agrees with the conclusion.
However, there are several issues need to improve. The reviewer would like suggests several issues as follows.
2) Specific comments for revision
- a)Major
#1 Are there any reports of testosterone changes in other enucleation procedures such as HoLEP?
This is the first manuscript for the effect post laser enucleation of prostate on testosterone level. Holmium laser enucleation of prostate (HOLEP) has multiple reports with testosterone, but they focus on the erectile function. They had the baseline testosterone level only without post op testosterone for comparison[1, 2].
There are some studies with transurethral resection of prostate (less resection ratio) published before (in our manuscript line 176-189). As our discussion, they had no testosterone change. We think it is due to enucleation could reach better clearance of the obstructed prostate, then less nocturia with better sleep quality make the elevated testosterone.
#2 How much did the frequency of nocturnal urination change before and after surgery in the patient group in this study? Is it possible that the decrease in nocturnal urination and the improved sleep environment may have increased postoperative testosterone?
Thanks for your comment. Indeed, our hypothesis is better sleep quality after prostate enucleation could restore the circadian rhythm and increase serum testosterone. We have added the nocturia before and after prostate enucleation in our manuscript in line 96-97, 112-116 and tables1, 2, 4
- b)Minor
#1 Low testosterone groups tended to have less prostatic nucleation weight. Is there any report that the larger the prostate volume, the lower the testosterone level?
Thanks for your comments. In our data the enucleated weight indeed looks less than the low testosterone group (18.6gm Vs 33.3gm). However, there is not statistically different between high and low testosterone groups (p =0.114). There is one report showing no association between testosterone and prostate volume[3].
Interrupted sleep would lower the testosterone level[4], and it is one of the multiple etiologies of testosterone deficiency. In our manuscript, we believe it is the un-interrupted sleep make the low testosterone group patient improved.
[1] Lee D, Yoo J, Sohn D, Yoon B. 022 Sleep Quality and Sexual Function Before and After Holmium Laser Enucleation of the Prostate. The Journal of Sexual Medicine. 2017;14:S10.
[2] Placer J, Salvador C, Planas J, Trilla E, Lorente D, Celma A, et al. Effects of holmium laser enucleation of the prostate on sexual function. Journal of Endourology. 2015;29:332-9.
[3] Kim WT, Yun SJ, Choi YD, Kim G-Y, Moon S-K, Choi YH, et al. Prostate size correlates with fasting blood glucose in non-diabetic benign prostatic hyperplasia patients with normal testosterone levels. Journal of Korean medical science. 2011;26:1214-8.
[4] Luboshitzky R, Zabari Z, Shen-Orr Z, Herer P, Lavie P. Disruption of the nocturnal testosterone rhythm by sleep fragmentation in normal men. The Journal of Clinical Endocrinology & Metabolism. 2001;86:1134-9.
Reviewer 3 Report
The current study aims to demonstrate a link between benign prostate hyperplasia enucleation and changes in serum testosterone levels.
Overall, the paper is well put together, with a thorough research protocol.
The introduction presents the current status of hypogonadism associated with the aging male syndrome, as well as the limitations of testosterone supplements. For this section, I would recommend that the authors include exact data and percentages of prostate cancer induced by testosterone supplements, as well as the proportion of patients with ameliorated lower urinary tract symptoms after chronic treatment with silodosin or vasopressin. Moreover, the working hypothesis is not clearly stated. I would advise the authors to describe the link and the pathophysiological mechanism between prostate adenoma enucleation and serum testosterone levels.
For Materials and methods section, I would suggest that the authors add the following data: how many surgeons performed the prostate enucleations and each surgeon’s experience, why transrectal ultrasound was chosen over multiparametric magnetic resonance imaging of the prostate and what was the therapeutic approach for patients with elevated PSA levels prior to the procedure, if a prostate biopsy has been performed in order to rule out prostate cancer.
The Results section could be further improved by analyzing if there are differences in terms of testosterone levels, lower urinary tract symptoms and postoperative complication rates when different enucleation techniques were employed (Thulium or Bipolar). Additionally, I would advise the authors to report the postoperative complication according to the Clavien-Dindo classification system.
Finally, the Discussions section presents the current literature data regarding how metabolic syndrome and diabetes mellitus affects testosterone levels, as well as other sex hormones. I would recommend for the authors to limit these topics to only one paragraph, as they do not analyze the link between obesity and hypogonadism, nor do they have data upon a complete metabolic profile. Furthermore, the authors should address and discuss their results more widely, compared with published papers that tackled similar research protocols.
Author Response
The current study aims to demonstrate a link between benign prostate hyperplasia enucleation and changes in serum testosterone levels.
Overall, the paper is well put together, with a thorough research protocol.
The introduction presents the current status of hypogonadism associated with the aging male syndrome, as well as the limitations of testosterone supplements. For this section, I would recommend that the authors include exact data and percentages of prostate cancer induced by testosterone supplements, as well as the proportion of patients with ameliorated lower urinary tract symptoms after chronic treatment with silodosin or vasopressin. Moreover, the working hypothesis is not clearly stated. I would advise the authors to describe the link and the pathophysiological mechanism between prostate adenoma enucleation and serum testosterone levels.
Q1. include exact data and percentages of prostate cancer induced by testosterone supplements, as well as the proportion of patients with ameliorated lower urinary tract symptoms after chronic treatment with silodosin or vasopressin
-à Currently, prostate cancer is still contraindicated in testosterone supplement package insert. However there is no evidence that testosterone would cause prostate cancer in “EAU guideline on Sexual and Reproductive Health”. And in recent 20 years, there are many trials for testosterone supplement in prostate cancer patients[1-4]. As a result, there would not be percentage of prostate cancer induced by testosterone supplement.
Silodosin and vasopressin improve the testosterone level by improving patients’ sleep quality and re-establishing their circadian rhythm. Lower urinary tract symptoms (LUTS) are one of the factors. We have revised the manuscript in line 50-51,54 to explain the pathophysiology of this testosterone elevation. And we have added preop/postop nocturia parameter for supporting this hypothesis in tables 1, 2, 4 and in manuscript line 96-97, 112-116.
Q2. Moreover, the working hypothesis is not clearly stated. I would advise the authors to describe the link and the pathophysiological mechanism between prostate adenoma enucleation and serum testosterone levels.
Silodosin and vasopressin improve the testosterone level by improving patients’ sleep quality and re-establishing their circadian rhythm. Prostate enucleation could decrease more nocturia and improve patient’s sleep quality and circadian rhythm. We have revised the manuscript on line 50-51,54 to explain the pathophysiology of this testosterone elevation.
For Materials and methods section, I would suggest that the authors add the following data: how many surgeons performed the prostate enucleations and each surgeon’s experience, why transrectal ultrasound was chosen over multiparametric magnetic resonance imaging of the prostate and what was the therapeutic approach for patients with elevated PSA levels prior to the procedure, if a prostate biopsy has been performed in order to rule out prostate cancer.
Q3. how many surgeons performed the prostate enucleations and each surgeon’s experience
We have single surgeon - Yu-Hsiang Lin. And he has around 100-150/year prostate enucleation done in recent 5 years. We have added single surgeon in our manuscript in line 66.
Q4. why transrectal ultrasound was chosen over multiparametric magnetic resonance imaging of the prostate and what was the therapeutic approach for patients with elevated PSA levels prior to the procedure, if a prostate biopsy has been performed in order to rule out prostate cancer
This is a retrospective study. As a result, we followed the clinical guideline performing digital rectal examination, PSA and prostate needle biopsy to rule out prostate cancer. The patients with prostate cancer would receive multiparametric MRI and bone scan for staging. The patients with prostate cancer receiving multiparametric MRI would not be included in this cohort. And we have our patients’ LUTS survey according to the European Association of Urology guideline “Management of Non-neurogenic Male LUTS” ~~~prostate imaging is performed by transabdominal (suprapubic) ultrasound or transrectal ultrasound in daily practice.
The Results section could be further improved by analyzing if there are differences in terms of testosterone levels, lower urinary tract symptoms and postoperative complication rates when different enucleation techniques were employed (Thulium or Bipolar). Additionally, I would advise the authors to report the postoperative complication according to the Clavien-Dindo classification system.
Q5. further improved by analyzing if there are differences in terms of testosterone levels, lower urinary tract symptoms and postoperative complication rates when different enucleation techniques were employed (Thulium or Bipolar).
Thanks for your comment. Thulium laser/Bipolar are different instruments/tools for performing the same enucleation technique. We have performed paired-T test based on different instrument (Bipolar/Thulium). There is no statistic difference in serum testosterone.
This manuscript focuses on the prostate enucleation and restoring the life/sleep quality and testosterone. Describing the thulium/bipolar complications/LUTS relieving would be better in another manuscript comparing the instrument effectiveness. Thanks for your comment, we would have another manuscript comparing different instruments in BPH surgery.
Q6. Additionally, I would advise the authors to report the postoperative complication according to the Clavien-Dindo classification system.
According to our hypothesis, it is the better sleep quality during the post op 1.5month makes the patients restore their circadian rhythm and increase serum testosterone. This is the reason why we have this parameter “Adverse events with ER visit”. These ER visits imply sleep disruption after discharge. Clavien-Dindo classification system evaluates the complication during admission period. This manuscript focuses on the serum testosterone level after operation, and ER visit would be better implying the life quality at home.
Finally, the Discussions section presents the current literature data regarding how metabolic syndrome and diabetes mellitus affects testosterone levels, as well as other sex hormones . I would recommend for the authors to limit these topics to only one paragraph, as they do not analyze the link between obesity and hypogonadism, nor do they have data upon a complete metabolic profile. Furthermore, the authors should address and discuss their results more widely, compared with published papers that tackled similar research protocols.
Q7. I would recommend for the authors to limit these topics to only one paragraph, as they do not analyze the link between obesity and hypogonadism, nor do they have data upon a complete metabolic profile. Furthermore, the authors should address and discuss their results more widely, compared with published papers that tackled similar research protocols
Before, there are two reports about the post transurethral resection of prostate(TURP) testosterone level as our manuscript reference list 20,21. And they found no testosterone elevation after operation. They compared the prolactin/LH change after operation. Maybe due to no statistical change of testosterone in previous TURP reports, the later prostate enucleation studies do not have testosterone change as an endpoint. Most of the post enucleation reports focus on sexual function ~“International Index of Erectile Function”(IIEF)[5, 6]. However, prostate enucleation could reach a higher resection ratio than TURP, then a better functional outcome with less nocturia could be achieved. As a result, this is the first manuscript for the effect of testosterone after laser enucleation of prostate. We have added the preop /postop nocturia parameter to further support this hypothesis.
[1] Agarwal PK, Oefelein MG. Testosterone replacement therapy after primary treatment for prostate cancer. The Journal of urology. 2005;173:533-6.
[2] Dupree JM, Langille GM, Khera M, Lipshultz LI. The safety of testosterone supplementation therapy in prostate cancer. Nature Reviews Urology. 2014;11:526-30.
[3] Gaylis FD, Lin DW, Ignatoff JM, Amling CL, Tutrone RF, Cosgrove DJ. Prostate cancer in men using testosterone supplementation. The Journal of urology. 2005;174:534-8.
[4] Isbarn H, Pinthus JH, Marks LS, Montorsi F, Morales A, Morgentaler A, et al. Testosterone and prostate cancer: revisiting old paradigms. European urology. 2009;56:48-56.
[5] Lee D, Yoo J, Sohn D, Yoon B. 022 Sleep Quality and Sexual Function Before and After Holmium Laser Enucleation of the Prostate. The Journal of Sexual Medicine. 2017;14:S10.
[6] Placer J, Salvador C, Planas J, Trilla E, Lorente D, Celma A, et al. Effects of holmium laser enucleation of the prostate on sexual function. Journal of Endourology. 2015;29:332-9.
Round 2
Reviewer 2 Report
The authors have satisfactorily addressed all my previous comments.
Reviewer 3 Report
I congratulate the authors for the revised version of their manuscript.
I think the current form is suitable for publication.